# Root System Response and Yield of Irrigated Rice in Relation to Irrigation, Potassium and Nitrogen under Subtropical Conditions

Md. Salahuddin Kaysar [1], Uttam Kumer Sarker [1], Sinthia Afsana Kheya [1], Ahmed Khairul Hasan [1], Md. Alamgir Hossain [2], Uzzal Somaddar [3], Gopal Saha [3], Apurbo Kumar Chaki [4,5], Abeer Hashem [6], Elsayed Fathi Abd_Allah [7] and Md. Romij Uddin [1,*]

[1]  Department of Agronomy, Bangladesh Agricultural University, Mymensingh 2202, Bangladesh; kaysar29@bau.edu.bd (M.S.K.); uttam@bau.edu.bd (U.K.S.); sinthia.agron@bau.edu.bd (S.A.K.); akhasan@bau.edu.bd (A.K.H.)

[2]  Department of Crop Botany, Bangladesh Agricultural University, Mymensingh 2202, Bangladesh; alamgircbot@bau.edu.bd

[3]  Department of Agronomy, Patuakhali Science and Technology University, Dumki, Patuakhali 8602, Bangladesh; uzzal04485@ag.pstu.ac.bd (U.S.); gopalagr@pstu.ac.bd (G.S.)

[4]  On Farm Research Division, Bangladesh Agricultural Research Institute, Gazipur 1701, Bangladesh; a.chaki@uq.net.au

[5]  School of Agriculture and Food Sciences, The University of Queensland, Brisbane, QLD 4072, Australia

[6]  Botany and Microbiology Department, College of Science, King Saud University, P.O. Box 2460, Riyadh 11451, Saudi Arabia; habeer@ksu.edu.sa

[7]  Plant Production Department, College of Food and Agricultural Sciences, King Saud University, P.O. Box 2460, Riyadh 11451, Saudi Arabia; eabdallah@ksu.edu.sa

*   Correspondence: romijagron@bau.edu.bd

**Abstract:** Irrigation and fertilizer are two essential factors affecting rice root traits and yield. In this respect, a pot experiment was performed at the *boro* (dry season irrigated) season of 2021–2022 in the Department of Agronomy of Bangladesh Agricultural University, Mymensingh, Bangladesh. The variety Binadhan-10 was planted at two irrigation conditions, i.e., saturation (S) and continuous flooding (CF); two potassium (K) doses, e.g., 65 kg ha$^{-1}$ (K$_{65}$) and 98 kg ha$^{-1}$ (K$_{98}$); and two nitrogen (N) doses, i.e., 140 kg ha$^{-1}$ (N$_{140}$) and 210 kg ha$^{-1}$ (N$_{210}$). The experiment was laid in a split plot design with eight treatments and replicated thrice. The findings confirmed the significant variation in irrigation, K and N and the effects on root number (RN), root length (RL), root volume (RV), leaf area index (LAI), total dry matter (TDM), yield attributes and yield. Considering the interaction among irrigation, K and N, the S conditions with K$_{65}$ and N$_{140}$ showed best performance in relation to root parameters. At 80 DAT, the highest RN (373.00), RL (1700.00 cm), RV (8.90 cm$^3$ hill$^{-1}$), LAI (4.94) and TDM (25.83 g plant$^{-1}$) was obtained from this combination. Grain yield (GY) and root traits, except root porosity, showed a significant positive association. Grain yield (GY) was the highest (27.12 g pot$^{-1}$) under S conditions with K$_{65}$ and N$_{140}$. Therefore, the variety Binadhan-10 can be successfully cultivated with K$_{65}$ and N$_{140}$ under S conditions.

**Keywords:** root volume; total dry matter; correlation matrix; rice; harvest index

## 1. Introduction

Rice (*Oryza sativa* L.) is a significant crop worldwide, and a crucial component of the daily nutrition for almost 3 billion people worldwide [1]. The basic functions of the roots of a plant are to absorb water and nutrients from the surrounding soil [2]. The roots play a crucial role in a variety of plant processes, such as the intake of water and nutrient ions, the generation of plant hormones, amino acids, and organic acids, and the provision of anchorage to the plants [3]. The physiology and structure of roots are closely connected

to the development and growth of aboveground parts of the plants [4]. Plant breeders are becoming more and more aware of the value of root development for preserving crop yields [5].

One of the most prevalent substances on earth, nitrogen (N), is crucial to contemporary agriculture [6]. N fertilizer has a substantial impact on root growth and Gaudin et al. [7] revealed that reduced amounts of N promoted root elongation. However, a rise in N levels was linked to an increase in biomass and root length, as discovered by Fan et al. [8]. It is believed that a lack of N fertilizer will lead to a low N concentration in the soil, which reduces N absorption by the plant and the accumulation of root and shoot biomass [9], and inhibits root extension. N deficiency in the roots might prevent root development, which would limit the growth of the shoots and reduce yield.

The most prevalent inorganic cation, essential for promoting healthy plant growth, is potassium (K) [10]. Numerous crucial enzymes, including those involved in the synthesis of protein, transportation of sugar, photosynthesis process and metabolism of C and N, are activated by K. For greater yield and quality, it is essential [11]. K in plants serves a number of purposes, including regulating the cell cycle, maintaining both root and shoot development, and carrying out cell death programs [12,13]. K shortages are often associated with slower root growth and a poorer ratio of root to shoot biomass [14,15], while in certain instances, the ratio of root and shoot remains stable or somewhat rises at varying K levels [16].

The formation of roots, and the viability and development of plants, are all greatly impacted by the moisture content of the soil [17]. Root development, water and nutrient intake and plant growth are all impacted by the soil's texture, which also impacts air and water circulation in the soil. We can learn a lot about water stress from studying rice roots, including how it occurs, how it is acquired, how to respond to it, and how to tolerate it [18]. Aerobic adaptation requires an understanding of how roots respond, particularly how effectively they absorb water [19]. Higher penetration, length of roots, and root to shoot weight ratio are root-related traits that make aerobic rice agriculture more able to adapt to water shortage conditions [20]. The development of crops depends on roots' capacity to uptake water as well as nutrients. Their importance is heightened in arid regions where plants must spread their roots into deeper soils to obtain the nutrients that are available in the wet soil because the topsoil is typically dry and nutrient-deficient. The relationship between crop yield and root biomass is often demonstrated to be significant and almost invariably linear [21].

Water has a direct impact on the soil nutrient availability needed for plant development and agricultural production. The physiological mechanisms of nutrient uptake by plants are probably determined by how nutrient interactions with water affect plant traits and, ultimately, crop development. Soil water deficiencies can, therefore, reduce nutrient transfer merely by limiting the amount of water that reaches the plant. The transpiration stream is the pathway through which minerals and other nutrients are moved from root to shoot [22]. Plant root development is intimately linked to soil variables such as moisture, oxygen, temperatures, and fertility, with moisture and the fertility being the two most important ones. These factors are also interdependent and interact with one another [23]. When fertilizers are given to the soil or substrate, water interacts with those nutrients in a way that either positively or negatively influences plant growth [24]. The significance of soil nutrients for plant development and agricultural output is directly correlated with the availability of water. Crops' capacity to obtain nutrients is considerably affected by water through (i) the transformation of nutrients to usable forms, (ii) the transportation of nutrient near to roots, and (iii) loss mechanisms [25]. Therefore, a relationship exists between water and nutrients regarding plant and root growth and yield. The majority of earlier research, however, did not take into account how water, N, and K interact to affect root development, plant growth, and yield. In light of this, the purpose of this research was to assess variations in root traits and grain yields of Binadhan-10 in relation to their interactions under a different irrigation regime with different K and N treatments.

## 2. Materials and Methods

### 2.1. Site and Plant Materials

In the net house of the Department of Agronomy, Bangladesh Agricultural University (latitude: 24°42′55″, longitude: 90°25′47″), the experiment was conducted in *boro* seasons of 2021–2022. The experimental site was in the Old Brahmaputra floodplain (AEZ-9) [26], with a subtropical monsoon climate with a humid environment. The variety Binadhan-10 was used as study materials. From the Bangladesh Institute of Nuclear Agriculture (BINA), inbred variety Binadhan-10's seeds were collected.

### 2.2. Experimental Design and Crop Management

The split plot design was used to perform this experiment. The irrigation treatments were $I_1$ saturation (S), $I_2$ continuous flooding (CF); the K treatments were 65 kg ha$^{-1}$ ($K_{65}$), 98 kg ha$^{-1}$ ($K_{98}$) and the N treatments were 140 kg ha$^{-1}$ ($N_{140}$) and 210 kg ha$^{-1}$ ($N_{210}$). Each pot (30 L plastic pots with 35 cm diameter) was put inside the net house with 25 kg of soil. The soil was collected from experimental field at 0–15 cm depth and it was clay–loam type with the following characteristics: pH–$H_2O$ 5.81, Ec (µs/cm) 138, organic carbon (%) 1.12, total N (%) 0.17, available P (ppm) 23.6, available K (ppm) 88.30, and available S (ppm) 58.70. The gathered soil was sun-dried, then crushed and blended well before being placed into the pots. First, a soil-filled pot was weighed, and then a porous pot was submerged during the night in a bowl of water to maintain saturation. Following that, the weight was taken, a computation for absorbing water was made, and it was treated as saturation. Before seedling transplantation, irrigation treatments were applied using drip irrigation method, They were kept up until harvest while regulating saturation levels on a daily weight basis (gravimetric method). For continuous flooding, irrigation was provided and plants were well-watered in the pot. Fertilizer doses (AEZ basis) for pot experiment were applied as 2.5 g, 2.81 g and 0.09 g pot$^{-1}$ as triple super phosphate (TSP), gypsum and zinc sulphate, respectively [27]. Muriate of potash (MoP) served as the source of K, $K_{65}$ (3.25 g MoP pot$^{-1}$) and $K_{98}$ (4.9 g MoP pot$^{-1}$), and was applied during the final pot preparation. The nitrogen supply was urea and N for $N_{140}$ (7.59 g urea pot$^{-1}$) and $N_{210}$ (11.39 g urea pot$^{-1}$). During the final pot preparation, one-third of urea and entire amounts of all other fertilizers were added. The rest of the urea was applied 20 and 40 days after transplanting (DAT), as the duration of the variety was relatively shorter (130 days). The Binadhan-10 seedlings, which had previously been raised in the seedbed, were transplanted into the pot after they had reached 40 days old. Occasionally, mostly in the early phases of development, weeds were noticed and removed. Notable insects and diseases were not found.

### 2.3. Determination of Root Morphological and Physiological Traits

Root morphological characteristics were noted at 20, 40, 60, 80 DAT and at harvest stage. About 3 plants were carefully removed from each pot using a deep dig to ensure that the main tap root and all lateral roots could be uprooted safely. The tested plants were kept in water-filled plastic bags for about 12 h. The roots were extensively cleaned using 1 mm mesh sieves in order to ensure that no root was left behind and to make easy root separation possible. The estimated value of various features was then averaged.

#### 2.3.1. Number of Root (RN)

The rice plants were gently uprooted after being watered. The roots were removed and then cleaned under running water. Every plant's RN was carefully counted manually with sufficient care and averaged.

#### 2.3.2. Root Length (RL, cm)

The length of the root was determined in centimeters from base to the tip of root using ruler, and the total of the measurements was calculated.

### 2.3.3. Root Volume (RV, cm3 hill$^{-1}$)

In order to measure root volume, the root masses were placed into a water-filled measuring cylinder. The increase in water level was measured and expressed as cm$^3$ hill$^{-1}$ [28].

### 2.3.4. Root Porosity (RP, %)

The stored roots were kept in water with airtight polybags to maintain their original temperature. Measurements were made of the pycnometer vials' weights with and without water. The sample roots were gently blotted using tissue paper. An analytical balance was used to calculate the root weight. The roots were inserted into a water-filled vial. In the event that air bubbles were found, they were expelled by gently moving the immersed roots with a clean needle within the pycnometer vial. An analytical balance was used to determine the weight of the water-filled pycnometer and fresh roots. After that, the roots were removed from the vial and blended in a glass mortar and pestle. The homogenate was transferred completely and filled the pycnometer to capacity. Weight was obtained after bringing the homogenate and pycnometer to room temperature. The following formula was used to calculate porosity [29]:

$$\% \text{ porosity} = \frac{W_{hr + w} - W_{fr + w}}{W_w + W_{fr} - W_{fr + w}} \tag{1}$$

Here, $W_{hr+w}$ = weight of homogenized roots and water-filled pycnometer vial, $W_{fr+w}$ = weight of fresh roots and water-filled pycnometer vial, $W_w$ = weight of water-filled pycnometer vial, $W_{fr}$ = weight of fresh roots.

### 2.3.5. Leaf Area Index (LAI)

The leaf area (LA) was calculated using a leaf area meter (LI 3100, Licor, Inc., LincoIn, NE, USA) after the leaf blades and sheaths were separated. The ratio of LA to ground area was calculated to be the leaf area index (LAI). The usual formulas were used to compute the LAI, crop growth rate (CGR), relative growth rate (RGR), and net assimilation rate (NAR) [30,31].

### 2.3.6. Total Dry Matter (TDM)

Three hills (plants) from each pot were uprooted at each development stage. Collected leaves, culms, and panicles of the samples were oven-dried in brown paper bags for 72 h at 65 °C before being weighed by an electronic balance to obtain the average data on their dry weights(ghill$^{-1}$). Summarizing the weights of dry plant parts yielded total dry matter (TDM).

### 2.3.7. Yield Attributes and Yield

At maturity (90% ripened grain), the entire plant was cut with a sickle at the ground level. The weight of the rice grains was determined and reported as g pot$^{-1}$ after adjusting for the 14% moisture content. Data for plant height (PH), no. of effective tiller plant$^{-1}$ (ET), length of panicle (PL), no. of grains panicle$^{-1}$ (GP), weight of 1000 grains (TGW), grain yield (GY) and straw yield (SY) for each plant were noted. The harvest index (HI, %) was calculated by dividing the grain biomass into the biological biomass of plant [32].

### *2.4. Root Cross-Section*

The cross-section of root was seen by cutting 2 cm of the root from the tip. The root was then preserved in water to keep it fresh and examined under microscope.

### *2.5. Statistical Analysis of Data*

The statistical program JMP Pro 16(SAS Institute Inc., Cary, NC, USA), was used to conduct the two-way analysis of variance (ANOVA) test. Tukey's honestly significant

difference (HSD) post hoc test was used to examine the mean differences at the 0.05 and 0.01 probability levels. The data visualization and correlation matrix were developed using R (R for Windows 4.1.2) and Sigma Plot v14 (Systat Software, Inc., San Jose, CA, USA, http://www.systatsoftware.com, accessed on 20 March 2023) [33].

## 3. Results

### 3.1. Morphological Traits of Root, Total Dry Matter and Leaf Area Index

The RN and RL were greatly influenced by the interaction between irrigation, K and N at all observations (Figure 1). At 80 DAT, in case of RN, the value ranged from 373.00 to 351.67. The highest value was found for the interaction between S condition, $K_{65}$ and $N_{140}$ (373.00), while the lowest was observed in interaction between CF, $K_{98}$ and $N_{210}$ (351.67) at 80 DAT. The highest value of RL was noticed in interaction among S condition, $K_{65}$ and $N_{140}$ (1700.00 cm) and the lowest was registered in interaction among CF, $K_{98}$ and $N_{210}$ (1655.75 cm) at 80 DAT.

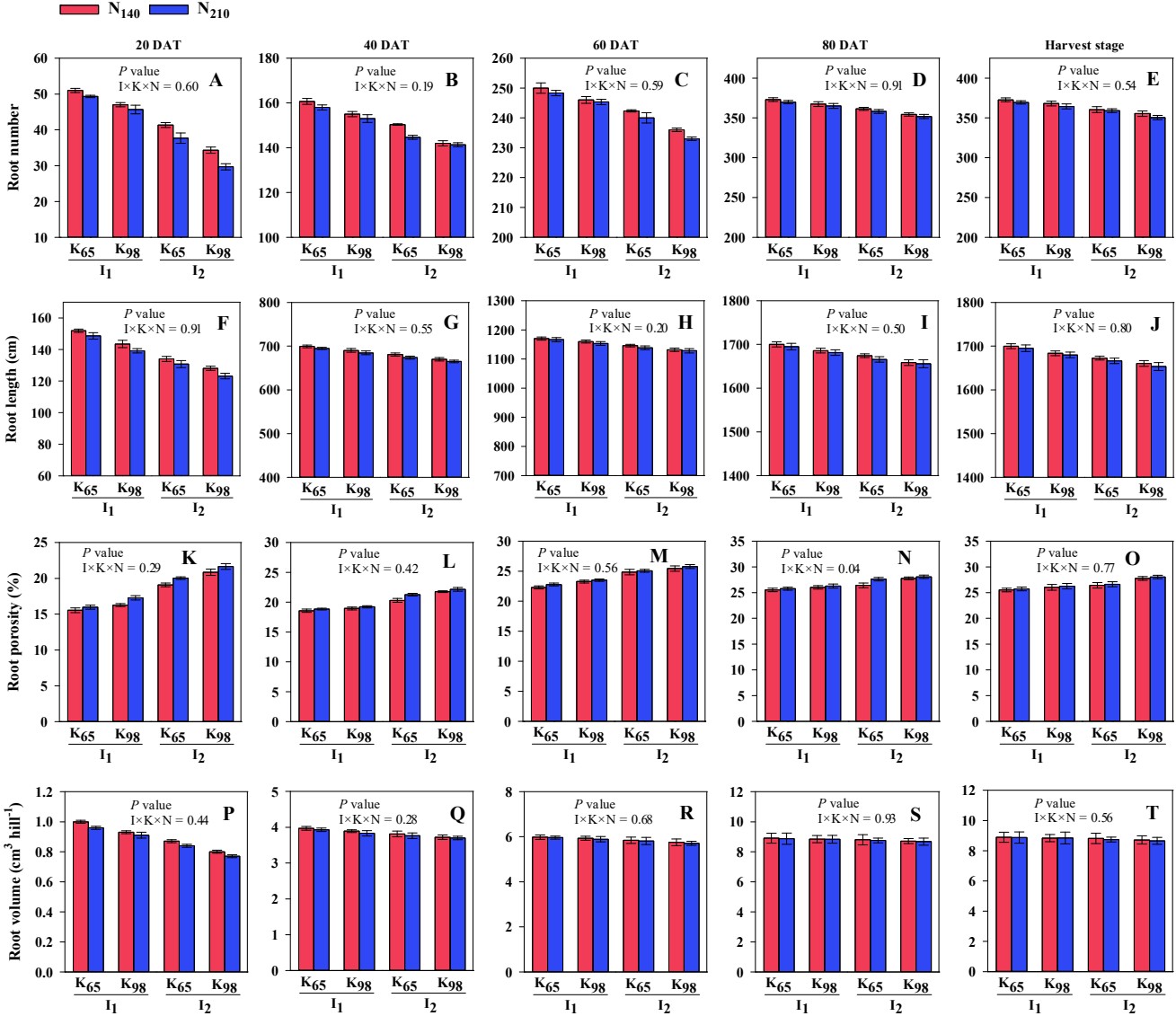

**Figure 1.** Dynamics of root morphological traits of Binadhan-10 under two irrigation, potassium and nitrogen treatments from 20 DAT to harvest stage. $I_1$: S $I_2$: CF $K_{65}$: 65 kg K ha$^{-1}$ $K_{98}$: 98 kg K ha$^{-1}$ $N_{140}$: 140 kg N ha$^{-1}$ $N_{210}$: 210 kg N ha$^{-1}$; (**A–E**) represent root number; (**F–J**) represent root length; (**K–O**) represent root porosity; (**P–T**) represent root volume.

The interaction among irrigation, K, and N had a substantial impact on RV and RP throughout all observations (Figure 1). In case of RV, the value ranged from 8.90 to 8.67 ($cm^3$ $hill^{-1}$) at 80 DAT. At 80 DAT, the highest value was found in the interactions among S condition, $K_{65}$ and $N_{140}$ (8.90 $cm^3$ $hill^{-1}$), while the lowest was observed in interactions among CF, $K_{98}$ and $N_{210}$ (8.67 $cm^3$ $hill^{-1}$). The maximum RP value was noticed in the interaction between CF, $K_{98}$ and $N_{210}$ (28.09%), and the lowest was registered in the interaction between S condition, $K_{65}$ and $N_{140}$ (25.52%) at 80 DAT.

The interaction among irrigation, K, and N had a substantial impact on the LAI as well. At 80 DAT, the highest (4.94) value of LAI was recorded under S conditions, along with $K_{65}$ and $N_{120}$ (Figure 2).

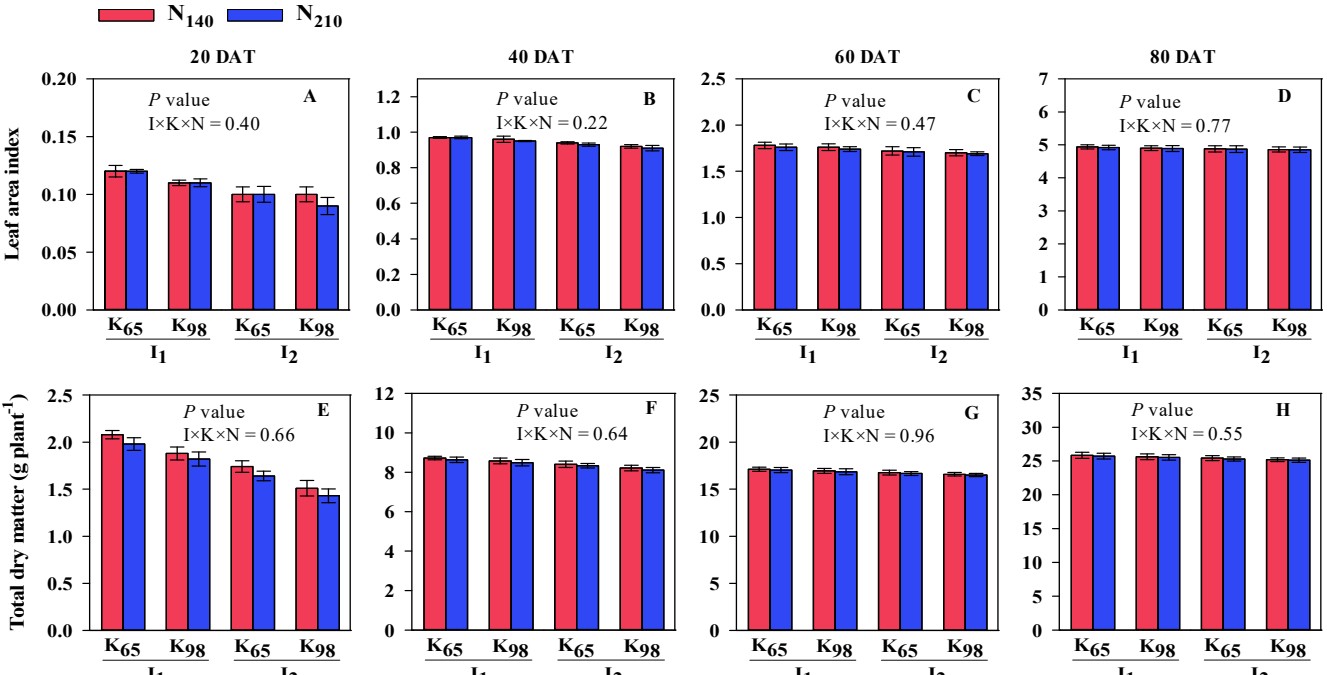

**Figure 2.** Leaf area index (LAI) and total dry matter (TDM) of Binadhan-10 under two irrigation, potassium and nitrogen treatments from 20 DAT to 80 DAT. $I_1$: S $I_2$: CF $K_{65}$: 65 kg $ha^{-1}$ $K_{98}$: 98 kg $ha^{-1}$$N_{140}$: 140 kg $ha^{-1}$ $N_{210}$: 210 kg $ha^{-1}$; (**A–D**) represent LAI; (**E–H**) represent TDM.

The combination of irrigation, K, and N had a considerable impact on TDM. In this situation, the highest (25.83 g $plant^{-1}$) value of TDM was found under S conditions, along with $K_{65}$ and $N_{140}$, whereas the lowest (25.11 g $plant^{-1}$) value was noticed in CF, along with $K_{98}$ and at $N_{210}$, at 80 DAT (Figure 2).

### 3.2. Growth Parameters

In all observations, the interaction of irrigation, K and N had a substantial impact on CGR, and the trend of CGR is linear with leaf area. At 60–80 DAT (3rd), the maximum (9.28 g $m^{-2}$ $day^{-1}$) CGR was registered with the S condition along with $K_{65}$ and $N_{98}$ (Figure 3).

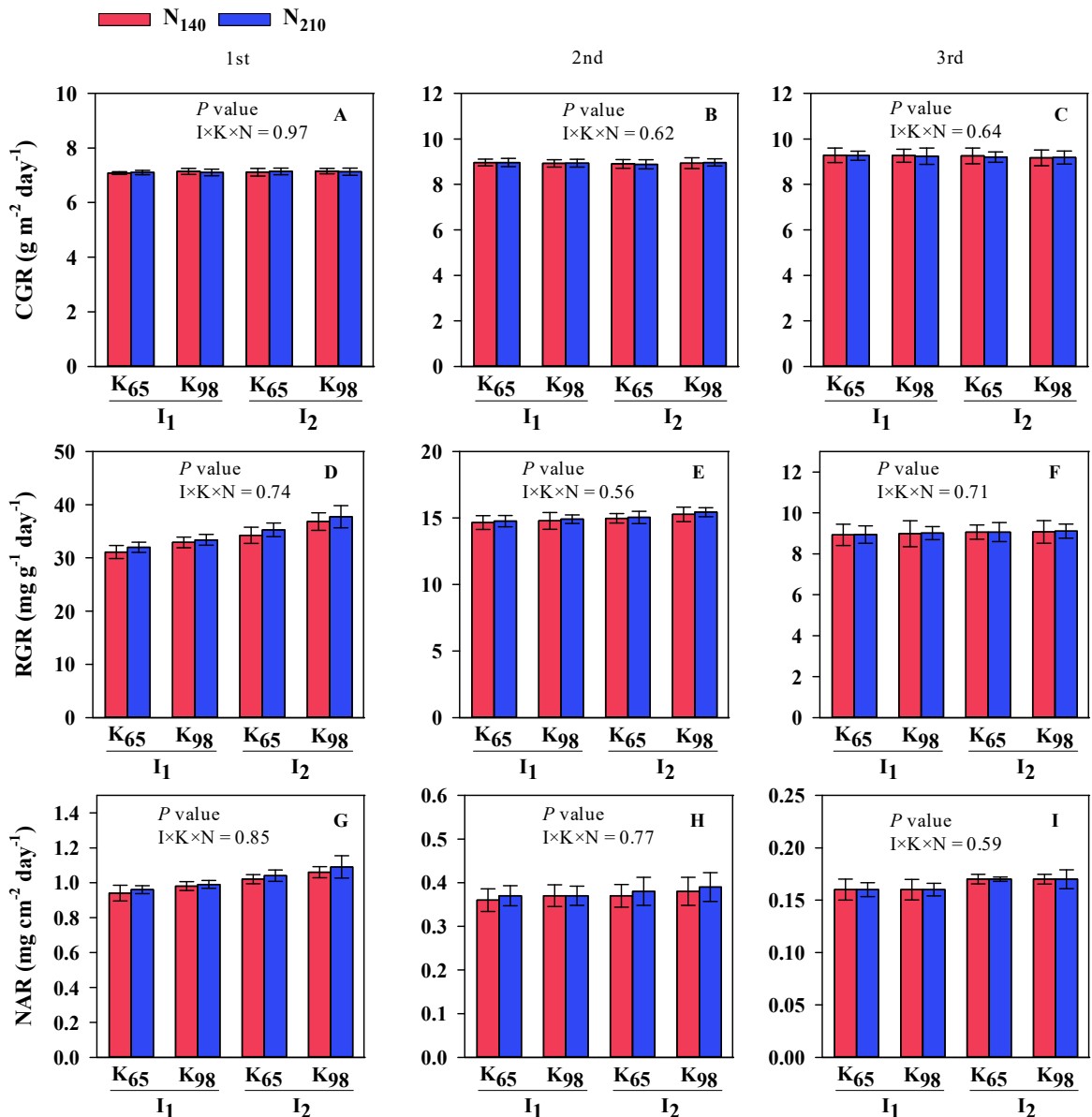

**Figure 3.** CGR, RGR and NAR of Binadhan-10 under two irrigation, potassium and nitrogen treatments at 20–40 (1st), 40–60 DAT (2nd) and 60–80 DAT (3rd). I1: S I2: CF $K_{65}$: 65 kg ha$^{-1}$ $K_8$: 98 kg ha$^{-1}$ $N_{140}$: 140 kg ha$^{-1}$ $N_{210}$: 210 kg ha$^{-1}$; (**A–C**) represent CGR; (**D–F**) represents RGR; (**G–I**) represent NAR.

The interaction between irrigation, K and N had a substantial impact on RGR. At 60–80 (3rd) DAT, the highest (9.11 mg g$^{-1}$ day$^{-1}$) RGR value was noticed in CF with $K_{98}$ and $N_{210}$, and the lowest (8.92 mg g$^{-1}$ day$^{-1}$) value was registered in S conditions with $K_{65}$ and $N_{140}$ (Figure 3).

At every observation, the interaction among irrigation, K and N had a considerable impact on NAR. The figure revealed that when LAI increased, NAR decreased in all interactions, which may be related to increased tillering and leaf area development. At 40–60 (2nd) DAT, the highest (0.39 mg cm$^{-2}$ day$^{-1}$) NAR value was found in CF with $K_{98}$ and $N_{210}$, and the lowest (0.36 mg cm$^{-2}$ day$^{-1}$) value was registered in S conditions with $K_{65}$ and $N_{140}$ (Figure 3).

### 3.3. Yield Contributing Parameters and Yield

Treatments with irrigation, K and N had a significant effect on rice yield components and output (Table 1). In irrigation, ET was higher (16.17) under S conditions than in CF (13.67). K revealed that, at $K_{65}$, the value was higher (15.67) than at $K_{98}$ (14.17). In case of N, the value was higher (15.33) at $N_{140}$ than at $N_{210}$ (14.50). A higher PL (24.29 cm) was found under S conditions than in CF (18.67 cm) in case of irrigation. Under K, at $K_{65}$ level, a higher value (22.50 cm) of PL was noticed than at $K_{98}$ (20.46 cm). On the other hand, at $N_{140}$, a higher PL (21.92 cm) was observed than at $N_{210}$ (21.04). In case of irrigation, the value of GP was higher (124.33) in S conditions than in CF (117.67). Similarly, in case of K, the higher value (122.83) was found at $K_{65}$ than at $K_{98}$ (119.17). Under N, the higher (121.75) value of GP was found at $N_{140}$ than at $N_{210}$ (120.25). In case of irrigation, the higher TGW (26.90 g) was noticed under S conditions compared to CF (22.81 g). In case of K, the higher (25.87 g) value was observed at $K_{65}$ compared to $K_{98}$ (23.85 g). N at $N_{140}$ produced a higher (121.75 g) value than at $N_{210}$ (120.25 g). In case of GY, under irrigation treatment, the value of grain yield was higher (25.70 g pot$^{-1}$) under S conditions than in CF (20.65 g pot$^{-1}$). In K, the value was higher (24.25 g pot$^{-1}$) at $K_{65}$ than at $K_{98}$ (22.10 g pot$^{-1}$). On the other hand, a higher (23.83 g pot$^{-1}$) value of GY was found at $N_{140}$ than at $N_{210}$ (22.52 g pot$^{-1}$).

**Table 1.** Yield components of Binadhan-10 under two irrigation, potassium and nitrogen treatments.

| Irrigation (I) | PH (cm) | ET (no.) | PL (cm) | GP (no.) | TGW (g) | GY (g pot$^{-1}$) | SY (g pot$^{-1}$) | HI (%) |
|---|---|---|---|---|---|---|---|---|
| $I_1$ | 96.33 a | 16.17 a | 24.29 a | 124.33 a | 26.90 a | 25.70 a | 26.05 a | 49.66 |
| $I_2$ | 91.58 b | 13.67 b | 18.67 b | 117.67 b | 22.81 b | 20.65 b | 20.98 b | 49.57 |
| CV (%) | 3.24 | 9.07 | 6.15 | 2.01 | 6.02 | 7.07 | 5.94 | 1.75 |
| Potassium (K) | | | | | | | | |
| $K_{65}$ | 95.33 a | 15.67 a | 22.50 a | 122.83 a | 25.87 a | 24.25 a | 24.49 a | 49.73 |
| $K_{98}$ | 92.58 b | 14.17 b | 20.46 b | 119.17 b | 23.85 b | 22.10 b | 22.54 b | 49.51 |
| CV (%) | 3.89 | 11.46 | 14.15 | 3.14 | 9.59 | 12.42 | 11.95 | 1.74 |
| Nitrogen (N) | | | | | | | | |
| $N_{140}$ | 94.67 a | 15.33 a | 21.92 a | 121.75 a | 121.75 a | 23.83 a | 24.14 a | 49.67 |
| $N_{210}$ | 93.25 b | 14.50 b | 21.04 b | 120.25 b | 120.25 b | 22.52 b | 22.89 b | 49.56 |
| CV (%) | 4.10 | 12.27 | 14.84 | 3.46 | 10.33 | 13.05 | 12.41 | 1.75 |
| ANOVA | | | | | | | | |
| I | ** | ** | ** | ** | ** | ** | ** | NS |
| K | * | ** | ** | ** | ** | ** | ** | NS |
| N | * | * | ** | * | * | ** | ** | NS |

Notes: Within every column, means indicated by the identical letters were not substantially dissimilar. **, * and NS denote significance at the 1%, 5% levels and non-significance, respectively, depending on the ANOVA.

The influence of interactions between irrigation, K and N treatments on yield and yield characteristics is shown in Figure 4. S conditions with $K_{65}$ and $N_{140}$ produced the highest (17.00) ET, while the lowest (12.00) ET was found at CF with $K_{98}$ and $N_{210}$. The highest value of PL (25.50 cm) was found in S conditions, along with $K_{65}$ and $N_{140}$, whereas the lowest (17.00 cm) was registered in CF with $K_{98}$ and $N_{210}$. The highest GP (126.67) value was found in interactions among S conditions, $K_{65}$ and $N_{140}$, whereas the lowest value was found in interactions among CF, $K_{98}$ and $N_{210}$ (115.00). In case of TGW, the highest (28.00 g) value was registered in S conditions with $K_{65}$ and $N_{140}$, while the lowest value (20.75 g) was observed in CF with $K_{98}$ and $N_{210}$. Finally, the highest (27.12 g pot$^{-1}$) value of GY was registered in S conditions along with $K_{65}$ and $N_{140}$, while the lowest (18.26 g pot$^{-1}$) value was noticed in CF along with $K_{98}$ and $N_{210}$.

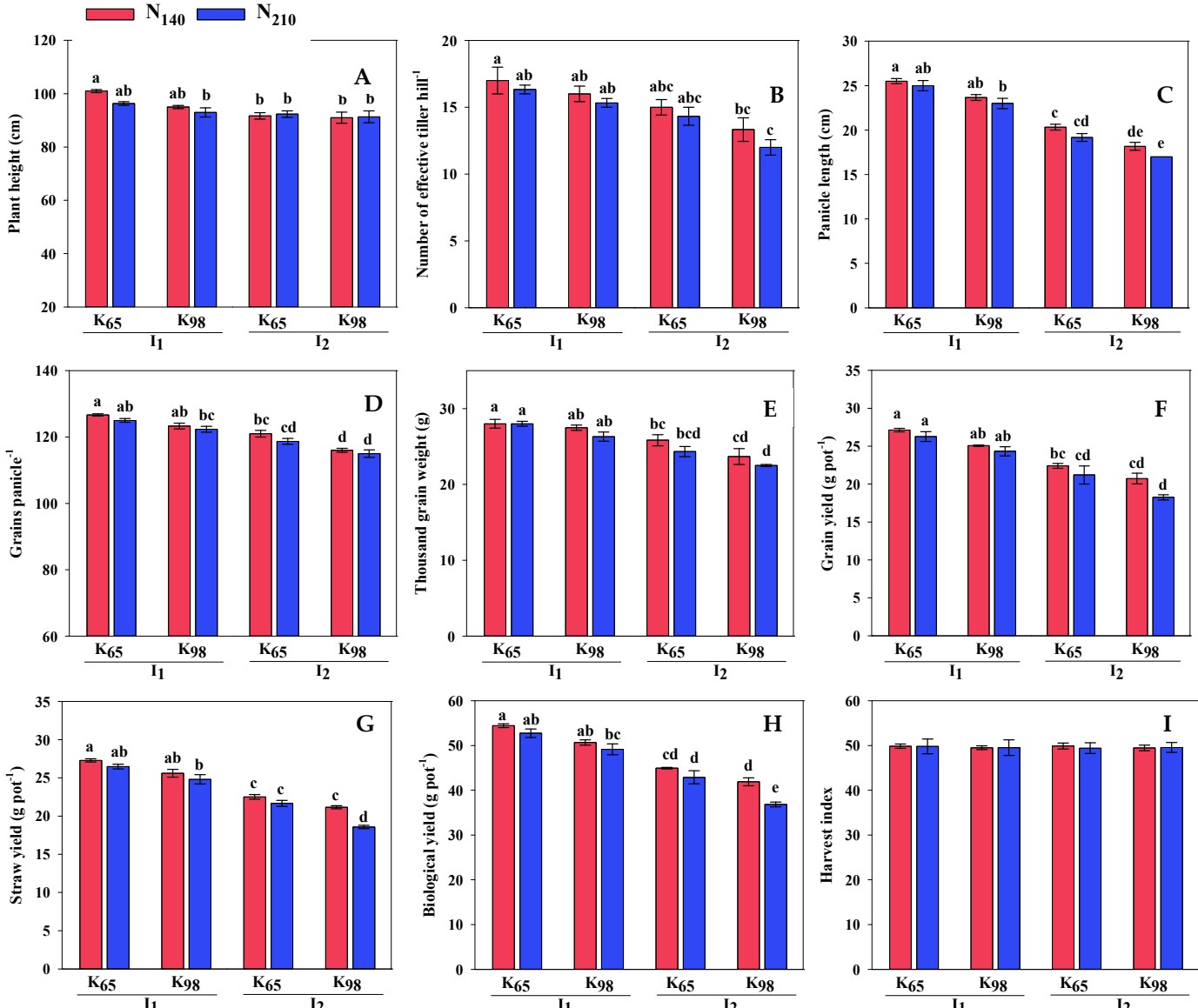

**Figure 4.** Yield and yield contributing parameters of Binadhan-10 under two nitrogen, potassium and irrigation treatments. The identical letters were not statistically dissimilar $I_1$: S $I_2$: CF $K_{65}$: 65 kg ha$^{-1}$ $K_8$: 98 kg ha$^{-1}$ $N_{140}$: 140 kg ha$^{-1}$ $N_{210}$: 210 kg ha$^{-1}$.

### 3.4. Relationship among Root Traits, Growth Indices, Yield and Yield Attributes

The correlation matrix of root traits, growth indices, yield, and yield parameters is shown in Figure 5 to examine their relationship. LAI had a significant and positive relationship with all root traits except RP. GY, SY and BY are significantly and positively correlated with RN, RL and RV, while they have a negative relationship with RP. Again, PH, ET, PL, GP, and TGW had significant relationships with RN, RL, and RV, while RP had relationships that were negative.

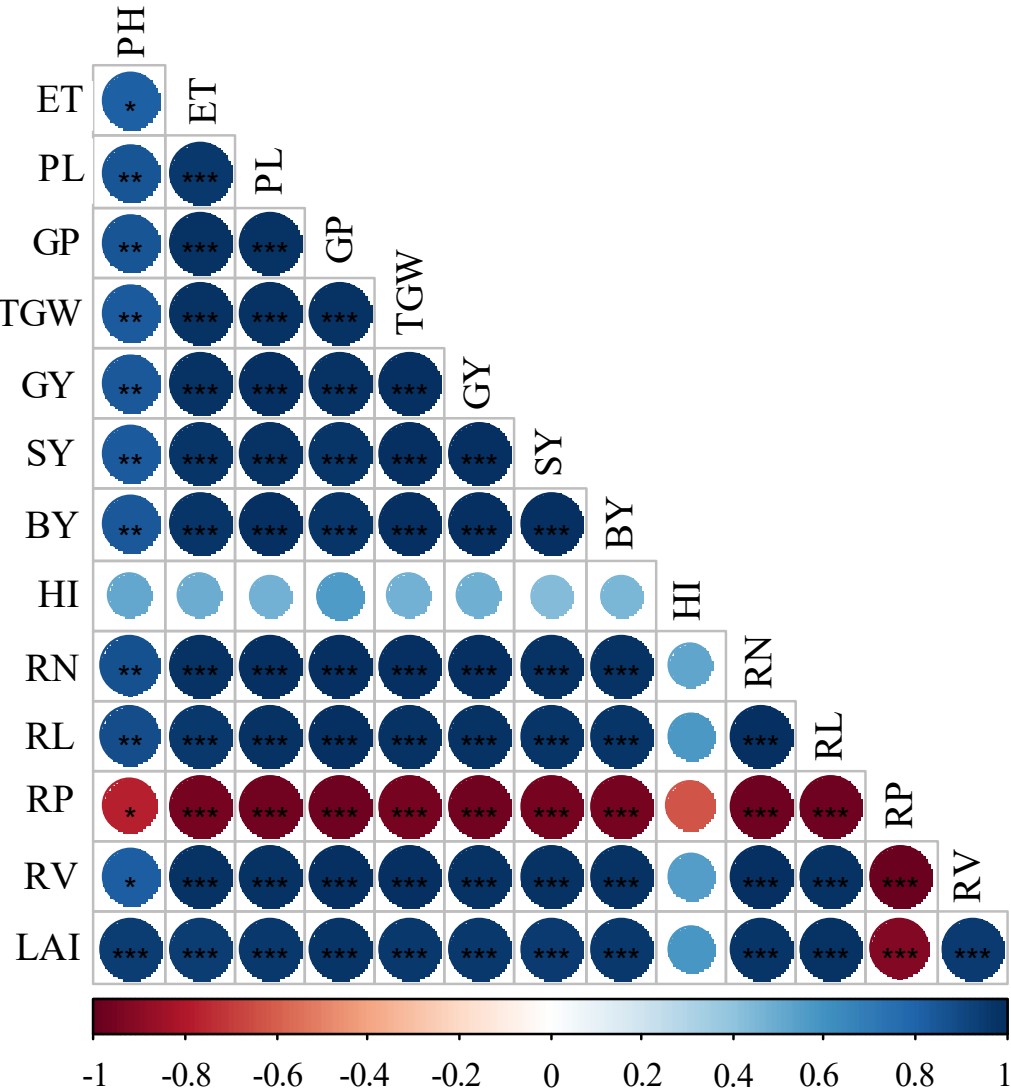

**Figure 5.** Correlation matrix and heatmap of the root traits, growth parameters, yield attributes and yield. The positive and negative correlations are indicated by blue and red ellipses. The greater coefficient is reflected by higher color intensity. *, ** and *** indicate level of significance at 5, 1 and 0.1% level of probability. Trait details: PH—plant height; ET—number of effective tillers hill$^{-1}$; RN—root number; RL—root length; RP—root porosity; RV—root volume; PL—panicle length; GP—grains per panicle; TGW—thousand grain weight; GY—grain yield; SY—straw yield; BY—biological yield; HI—harvest index.

### 3.5. Root Cross-Sectional View

The root cross-section of Binadhan-10 at 80 DAT under different combination of treatments is shown in Figure 6. In case of S conditions with K and N treatments, no aerenchymatous tissue was found. In the combination of CF with K and N treatments, aerenchymatous tissue was developed. From the cross-section, it was found that, under CF at higher dose of K and N, that is, at $K_{98}$ and at $N_{210}$, the formation of aerenchyma was greater than at $K_{65}$ and $N_{140}$ levels.

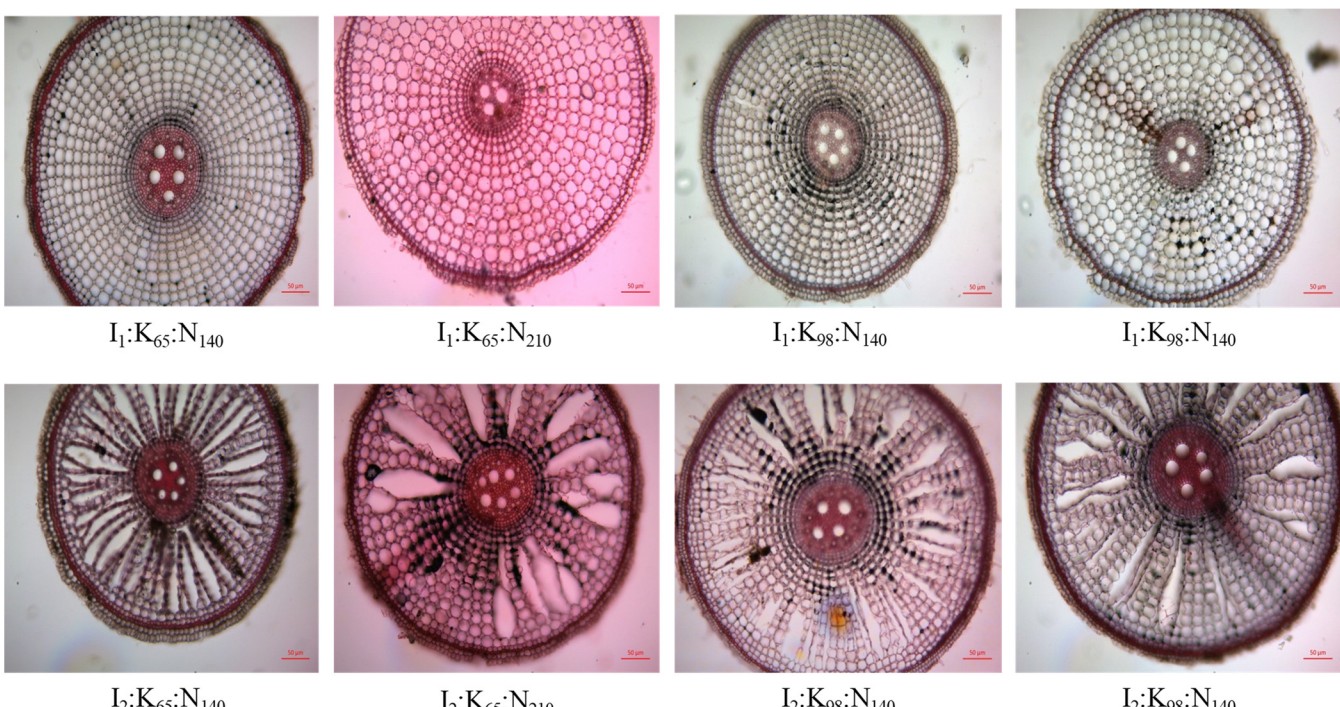

| $I_1$:$K_{65}$:$N_{140}$ | $I_1$:$K_{65}$:$N_{210}$ | $I_1$:$K_{98}$:$N_{140}$ | $I_1$:$K_{98}$:$N_{140}$ |

| $I_2$:$K_{65}$:$N_{140}$ | $I_2$:$K_{65}$:$N_{210}$ | $I_2$:$K_{98}$:$N_{140}$ | $I_2$:$K_{98}$:$N_{140}$ |

**Figure 6.** The differences in root cross-sections of Binadhan-10 under different combinations of treatments at 80 DAT. Here, $N_{140}$: 140 kg ha$^{-1}$, $N_{210}$: 210 kg ha$^{-1}$, $K_{65}$: 65 kg ha$^{-1}$, $K_{98}$: 96 kg ha$^{-1}$; $I_1$: saturation $I_2$: continuous flooding.

## 4. Discussion

The research has investigated the solo impact of irrigation, potassium and nitrogen on rice roots. However, research on the interaction between irrigation, potassium and nitrogen, and the effect on rice root, is still limited. In this experiment, root traits, yield and yield-contributing features of Binadhan-10 varied under interactions with different irrigation treatments, K and N. This study set out to study changes in root traits and how they related to GY while subjected to various irrigation treatments, K and N.

An essential agronomic indicator, the LAI, measures crop growth and forecasts crop production. Crop yields are significantly influenced by leaf area [34]. A proper LAI is a key indicator of great crop output, balancing the growth of each organ in crops and regulating the source–sink relationship of crops. The growth factors that improve the yield of rice include CGR, RGR and NAR. The variety and growth stage have a significant impact on these growth factors. These growth metrics were significantly impacted by irrigation, K and N in this research. The LAI, CGR, RGR and NAR increased markedly under S conditions, at $K_{65}$ and at $N_{140}$ compared to CF, $K_{98}$ and $N_{210}$. The term "interactions between water and fertilizers" describes how water and the nutrients added to the soil interact with each other, having a favorable impact on plant growth [24,35–37]. The interaction between water and nutrients had a considerable and advantageous impact on the growth indices of the rice plant, according to our findings. Our results corroborated the claims made by Fageria et al. [38] that rice growth can be accelerated by the N and K interactions. The availability of one nutrient affects how well another is absorbed and interactions between nutrients are governed. Since $K^+$ functions as an electrochemical balance for $NO^{3-}$, applying K can increase plants' ability to absorb N [39]. N availability is affected by the interactions between $K^+$ and $NH^{4+}$ during this exchange [40]. In appropriate soil moisture conditions, N and K interactions accelerated growth indices in this research. Nutritionally, $K^+$ and $NH^{4+}$ have an antagonistic connection; nevertheless, $K^+$ and $NO^{3-}$ acquisition rates are frequently observed to be favorably associated [41], and an adequate K supply can boost amino acid and protein synthesis, stimulate N metabolism, and enhance rice plant growth and production [42].

In this study, the value of RN, RL and RV was higher under S conditions than CF, which followed the theme that irrigation that conserves water is more successful than flooding irrigation at enhancing root activity and establishing a healthy root morphology [43,44]. K has a variety of activities in plants, including controlling the cell cycle [12] and carrying out cell death programs, which both contribute to supporting root development [13], and that is why in our research at the $K_{65}$ (standard dose) level, the highest value of root traits was found, as opposed to the $K_{98}$ level. Furthermore, an overdose of N is detrimental to root growth, which is connected to the statement of Britto and Kronzucker [45] that excess N fertilizer might have a detrimental impact on root development due to ammonium toxicity and, under the best conditions, N fertilization increases root length and diameter [46]. The consequence of an optimal supply of moisture and N is enhanced root growth, as previously documented by Mahajan et al. [47]. However, water and N interact and have a linked influence on root development [48]. Again, root development and the growth of plants can be influenced by the input of N and K fertilizers or the combination of these fertilizers [49–52]. Combining K and N applications produced a notable beneficial reciprocal effect on crops and was a key strategy for increasing K usage efficiency [53]. According to studies, increased root length and root biomass were related to better N absorption through nutrient interactions such as N and K [54,55]. The findings of this study showed that root properties were enhanced by interactions between irrigation, K and N. As a result, we hypothesized that a balanced fertilization and optimal irrigation supply might enhance root development, growth indices, and ultimately crop production. This research furthered our knowledge of the relevance of balanced fertilization and the ideal watering conditions for enhanced root characteristics and can offer practical advice for the optimization of fertilizer management based on the root development response to nutrient supply.

There have been several publications on the connections between high yield and the accumulation and translocation of dry materials. The quantity of carbohydrates that plants store before heading and those that plants create through photosynthesis after heading are the primary determinants of the GY, which is a byproduct of dry matter production [56]. One of the most crucial plant nutrients, N, is essential for the creation of biomass and photosynthesis in plants. The primary contributing component to the increase in yield after N incorporation is an increase in panicle numbers [57,58]. In this research, a high RP was observed in CF and anaerobic conditions as a result of aerenchyma development (Figure 5), agreeing with the statement of Lynch et al. [59] that increased aerenchyma production is a root response to low oxygen levels. The aerenchyma's ability to transfer oxygen helps rice roots in submerged soils meet their oxygen requirements. Nutritional imbalanced in the rhizosphere result in the induction of aerenchymatous tissue [60,61]. Moreover, K, in particular, can enhance chlorophyll, shield the photosynthetic organs from dryness, and enable them to properly perform their function, which will boost photosynthesis. Together, these factors enhance the output of dry land crops [53]. Therefore, an optimum supply of N and K under optimum irrigation conditions can increase TDM, ET and GP.

Rice output is thought to be increased by a synergistic relationship between soil moisture and nitrogenous fertilizer during rice growth [62,63]. According to prior research by Mandal et al. [64], this may be attributed to a constant and optimal supply of N being present, together with optimal soil moisture in the root zone during the crop growth cycle. The application of K and N enhanced plant K content, GP, and straw production [65]. The grain yield may be impacted by the interactions between potassium (K) and nitrogen (N), according to the studies of Duncan et al. [55]. As per Metho et al. [66], with the combined application of N and K fertilizers, the yield was higher than when the nutrients were added separately. In the correlation matrix, the value of RL, RN and RV was significantly correlated with the GY, along with interactions between K and N. According to prior research, N and K showed the significant influence of RL, RV, and RN in relation to GY [51,55,67], which may be due to the optimum supply of irrigation that can promote the activity of nutrients.

## 5. Conclusions

Based on this study, an optimum supply of water and dose of potassium and nitrogen positively influences root traits, growth indices and yield. At S with $K_{65}$ and $N_{140}$ levels, the value of RN, RL, RV, LAI TDM, yield attributes and yield peaked, followed by continuous flooding, $K_{98}$ and $N_{210}$ levels. The highest values of RN, RL and RV were found at 80 DAT; then, these values were decreased at the harvest stage in most cases. A positive correlation was found between root traits and yield, except root porosity. It can be concluded that interactions between irrigation, potassium and nitrogen are significantly beneficial for rice root traits, as well as growth indices and yield.

**Author Contributions:** Conceptualization: M.S.K., U.K.S., M.R.U., A.K.H. and M.A.H.; methodology, data collection and original data analysis: M.S.K., S.A.K. and U.S.; data presentation, writing: M.S.K., A.H. and E.F.A.; reviewing and editing: U.K.S., M.R.U., M.A.H., E.F.A., G.S. and A.K.C.; funding acquisition: M.R.U. and E.F.A. All authors have read and agreed to the published version of the manuscript.

**Funding:** This research was funded by Bangladesh Agricultural University Research Systems (BAU-RES) Project number (2019/15/BAU), Bangladesh. The authors would like to extend their sincere appreciation to the Researchers Supporting Project Number (RSP2023R134), King Saud University, Riyadh, Saudi Arabia.

**Data Availability Statement:** Data sets analyzed during the present study are accessible from the current author on reasonable request.

**Acknowledgments:** The authors extend their appreciation to BAURES Project number (2019/15/BAU), Bangladesh Agricultural University, Mymensingh, The authors would like to extend their heartfelt gratitude to the Researchers Supporting Project Number (RSP2023R134), King Saud University, Riyadh, Saudi Arabia.

**Conflicts of Interest:** The authors state no conflict of interest.

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
