# Peer review of "Root System Response and Yield of Irrigated Rice in Relation to Irrigation, Potassium and Nitrogen under Subtropical Conditions"

_agronomy, doi:10.3390/agronomy13061626_

Round 1

Reviewer 1 Report

This study investigated the effectiveness of irrigation, K and N on root number (RN), root length (RL), root volume (RV), leaf area index (LAI), total dry matter (TDM), yield attributes and yield. There is much data and results show support for the hypothesis presented in a certain. But there still remain some errors or confuse in the manuscript.

Writing:

1)      Authors should be more to emphasize manuscript novelty. The English used in this manuscript is simply too confusing. Verb tenses and meanings are mixed, which makes for ambiguous sentences and truncated reading.

    Eg: Line 30 yield attributes and line 176 yield parameters. Is it necessary to unify?

2)      The manuscript contains several writing errors that need to be corrected. Here I am mark up some of these errors, as most can be found with adequate proofreading.

Eg:(1) Please correct "parameters of Bina dham-10" in line 315 to "parameters of Bina dham-10".

(2) Please correct "withRP" in line 322 to "with RP".

(3) Many references such as Line 502, line 507 are not followed by DOI. Please ensure that the reference format is uniform.

(4) Line 469, line 583, line 613: Zea mays, Oryza sativa needs italics.

(5) Line 536(2006) needs to be deleted.

(6) Line 457 Please confirm whether [J] needs to be kept. If you need to keep it, please make sure that the format is uniform.

3) The materials and methods section at times reads like a laboratory protocol. It is great when authors include details that can increase reproducibility, but when the text is as confusing as this, that is not accomplished.

Eg: How to measure RN, RL, whether instruments are used, and what is the instrument model.

4) Acronyms are used indiscriminately. Some are not defined at first mention; some are defined and then never again used; most are outright unnecessary. Eg: I1-Saturation (S), I2- Continuous flooding (CF). I1 and I2 only appear in the figure and are never mentioned in this article.

Figures:

1)      The figures are aesthetically pleasing, although authors should be mindful of colour schemes and of how presenting all data is not always the best way to convey information visually.

Eg: (1) Different pictures have different fonts. The font used in Figure 5 is Arial, while the font used in Figure 4 is Times New Rome.

   (2) Fonts in pictures are distorted when added. Please further beautify Figure 5 and 6.

2) The presentation of picture data is incomplete.

    Eg: The vertical coordinate values in Figure 4 are incomplete.

The quality of English language should be improved.

Author Response

Dear Reviewer,

Thank you for your valuable and logical comments about the manuscript. We already addressed your comments and make necessary corrections. We believe after corrections, the quality of manuscript is definitely improved. The responses are as follows.   

Point 1:    Authors should be more to emphasize manuscript novelty. The English used in this manuscript is simply too confusing. Verb tenses and meanings are mixed, which makes for ambiguous sentences and truncated reading.

    Eg: Line 30 yield attributes and line 176 yield parameters. Is it necessary to unify?

Response 1: It’s a very useful comment for improving the manuscript. We have unified in line 30 and line 176 as yield attributes.

Point 2. Eg:(1) Please correct "parameters of Bina dham-10" in line 315 to "parameters of Bina dham-10".

Response 2: It’s a very appreciable comment. It has been corrected accordingly.

Point 3. Please correct "withRP" in line 322 to "with RP".

Response 3: Thank you for nice comment. It has been done.

Point 4: Many references such as Line 502, line 507 are not followed by DOI. Please ensure that the reference format is uniform.

Response 4: We certainly appreciate your useful comment. We have rigorously search for all DOI and added in references 22, 29 and 30.

Point 5: Line 469, line 583, line 613: Zea mays, Oryza sativa needs italics.

Response 5: Thank you for your nice comment indeed. These are already corrected.

Point 6. Line 536:(2006) needs to be deleted.

Response 6: It has already been done accordingly.

Point 7. Line 457: Please confirm whether [J] needs to be kept. If you need to keep it, please make sure that the format is uniform.

Response 7: Thank you for valid comment. The mentioned letter has been deleted for uniform format.

Point 8. The materials and methods section at times reads like a laboratory protocol. It is great when authors include details that can increase reproducibility, but when the text is as confusing as this, that is not accomplished.

Eg: How to measure RN, RL, whether instruments are used, and what is the instrument model.

Response 8: Thank you for nice comment. The measurement of RN, RL has been clarified more for understanding.

Point 9. Acronyms are used indiscriminately. Some are not defined at first mention; some are defined and then never again used; most are outright unnecessary. Eg: I1-Saturation (S), I2- Continuous flooding (CF). I1 and I2 only appear in the figure and are never mentioned in this article.

Response 9. Thank you for effective comment. We checked and used abbreviated form in material and methods for I1-Saturation (S), I2- Continuous flooding (CF) (2.2) and used these later different figures. For others, checked accordingly.

Point 10.  Different pictures have different fonts. The font used in Figure 5 is Arial, while the font used in Figure 4 is Times New Rome.

Response 10. Thank you for legal comment. The font has been changed and now both are Times New Romans

Point 11. Fonts in pictures are distorted when added. Please further beautify Figure 5 and 6.

Response 11. Thank you very much. These has been changed and now ok.

Point 12.   The vertical coordinate values in Figure 4 are incomplete.

Response 12. Thank you so much. The vertical coordinate are complete now.

Reviewer 2 Report

The paper aims to investigate the response of rice root systems to irrigation, potassium, and nitrogen. The regulation of irrigation, potassium, and nitrogen on rice roots, plant growth, and yield was documented, and the optimal treatment of irrigation, K, and N was developed. My concerns are as follows:

1, Abstract: How many treatments are in this study? It should be stated clearly.

2. Lines 29-31: Is the result significant?

3. Lines 31-32: what's the best performance?

4. Abstract: what's the conclusion?

5. Line 125: what/s the duration of the rice season?

6. Statistical results should be shown in Figures 1-3.

7. Lines 346-349: There were no results and discussion on the interactive effects of irrigation, K, and N on rice root and yield. 

The English language is OK

Author Response

Dear Reviewer,

Thank a lot for your appreciated and rational comments regarding the manuscript. The response are listed below.

       Point 1. Abstract: How many treatments are in this study? It should be stated clearly.

      Response 1: Thank you for valid comment. It has been clearly stated in abstract.

      Point 2. Lines 29-31: Is the result significant?

      Response 2. This result is significant and mentioned correctly.

Point 3. Lines 31-32: what's the best performance?

Response 3. It has been stated correctly.

Point 4. Abstract: what's the conclusion?

Response 4. Thank you for nice comment. The conclusion has been added accordingly.

Point 5. Line 125: what/s the duration of the rice season?

Response 5: It is a very appreciable comment. The duration has been added in the mentioned line.

Point 6. Statistical results should be shown in Figures 1-3.

Response 6: Thank you for valid comment. Statistical results are now added in Fig. 1-3.

Point 7. Lines 346-349: There were no results and discussion on the interactive effects of irrigation, K, and N on rice root and yield. 

Response 7: Thank you nice comment. Actually we searched a lots of literature and previous research work but unfortunately no research was found on interaction of irrigation, K, and N on rice root and yield. We think, this is the novelty of our research in this respect.

Point 8. Comments on the Quality of English Language

The English language is OK

      Response 8: Thank you for appreciable comment.

Round 2

Reviewer 2 Report

The authors have improved the manuscript substantially according to the comments and suggestions of reviewers.  There are still several minor points that should be addressed before accepting for publication.

1. Keywords: change 'cross sectional view' to 'rice'.

2. Line 114: Where the soil in the pots was taken? The physical and chemical properties of the soil should be shown.

3. In the section 2.2, the management of irrigation and drainage should stated clearly.
